# REPO: DETOXIFYING LLMS VIA REPRESENTATION ERASURE-BASED PREFERENCE OPTIMIZATION

## ABSTRACT

Large language models (LLMs) trained on web-scale data can produce toxic outputs, raising concerns for safe deployment. Prior defenses, based on applications of DPO, NPO, and similar algorithms, reduce the likelihood of harmful continuations, but not robustly so: they are vulnerable to adversarial prompting and easily undone by fine-tuning–based relearning attacks. Indeed, research has shown that these edits to the model are superficial: linear probing reveals that harmful "directions" remain present in representations. Motivated by these findings, we propose Representation Erasure-based Preference Optimization method (REPO), which builds on SURE (Sepahvand et al., 2025), an unlearning algorithm originally developed for classification. Our core strategy is to preserve the representations of benign (safe, nontoxic) generations while forcing the representations of toxic generations to converge toward their benign counterparts. This alignment is achieved through a coupled objective, which combines a retain loss on non-toxic samples with a domain-adversarial loss on both toxic and non-toxic samples, enforced by a gradient reversal layer. Comprehensive evaluations show that REPO not only significantly reduces in-distribution and out-of-distribution toxicity compared to baselines like DPO, NPO, and RMU, but also achieves best-in-class robustness against sophisticated attacks, including relearning on forget and retain samples, and adversarial prompt injection, via an enhanced variant of GCG.

## 1 INTRODUCTION

LLMs trained on massive, uncurated corpora can exhibit undesirable behaviors, including the memorization and regurgitation of hazardous knowledge (Li et al., 2024), the generation of toxic language (Wen et al., 2023), and the amplification of social biases engrained in large-scale web data (Sheng et al., 2019; Gehman et al., 2020). These risks have motivated the development of alignment algorithms aimed at reducing these behaviors. However, the application of existing alignment algorithms has proven fragile: while they can mitigate such behaviors, models remain vulnerable to jailbreak attacks that bypass safeguards and elicit harmful generations (Singh et al., 2025; Schwinn et al., 2024). For example, Greedy Coordinate Gradient (GCG), an adversarial attack that appends optimized suffixes to harmful queries, achieves high jailbreak success rates in eliciting harmful outputs across a variety of aligned models (Zou, Wang, et al., 2023; Jia et al., 2024).

Unlearning has emerged as a complementary strategy for mitigating problematic content in pretraining data, including private information and toxic language (Liu et al., 2025; Xu et al., 2023). Unlike safety training approaches that suppress harmful outputs, unlearning aims to removing hazardous capabilities from models altogether, making them inaccessible even to adversaries (Singh et al., 2025; Liu et al., 2025) with white or blackbox access. Early results indicate that unlearning can be effective against certain attacks; for example, methods such as RMU (Li et al., 2024) were observed to be resistant to linear probing of internal activations (Burns et al., 2023) and to classic forms of adversarial prompting like GCG (Zou, Wang, et al., 2023; Huu-Tien et al., 2025). Latent adversarial training can strengthen robustness by perturbing intermediate activations, thereby suppressing undesirable behaviors (Sheshadri et al., 2024). However, unlearning is not a panacea, as more adaptive jailbreaks have been shown to succeed (Łucki et al., 2025; Singh et al., 2025; Hu, Fu, et al., 2024). Among the most effective attacks are relearning attacks, which recover supposedly removed capabilities through lightweight fine-tuning on as few as ten unrelated examples (Hu, Fu, et al., 2024),

Figure 1: A schematic representation of REPO. Its regressor can be attached to any transformer block $M$ targeted for unlearning; here, $M$ is taken as the final transformer block before the linear unembedding layer. For each prompt, the retain (nontoxic) continuation $x_r$ and the forget (toxic) continuation $x_f$ are fed into the network, and the discriminator is trained to distinguish between toxic and nontoxic inputs.

and enhanced versions of GCG, which substantially improve attack success against RMU and NPO with only small modifications to the original GCG loss function (Łucki et al., 2025).

Motivated by these vulnerabilities, recent work has increasingly explored representation-based approaches that intervene directly on hidden representations rather than on the outputs of the model. Embedding-based unlearning, for instance, has been shown to be more resilient to paraphrasing attacks, preventing forgotten knowledge from resurfacing under semantic variations of prompts (Spohn et al., 2025). Mechanistic localization of unlearning to factual recall pathways likewise improves robustness against relearning by preventing capabilities from being restored through lightweight fine-tuning (Guo et al., 2025). Representation-level interventions also resist membership inference and inversion attacks, offering stronger privacy guarantees for forgotten data (Hu, Huang, et al., 2025). Overall, these results suggest that targeting hidden features can enable more durable forgetting, improving stability and resistance to knowledge recovery compared to output-level or gradient-based approaches (Muhamed et al., 2025; Jung et al., 2025; Wang, Li, et al., 2025).

Building on these findings, we propose the Representation Erasure-based Preference Optimization (REPO) method to detoxify large language models. REPO adapts the Selective Unlearning via Representation Erasure (SURE) method (Sepahvand et al., 2025), which was originally developed for the classification setting to erase the influence of a "forget set" by adjusting a model's representations. In our adaptation, REPO aligns the representations of toxic continuations with those of non-toxic ones, which reduces the model's tendency to exhibit toxic behavior while preserving its ability to generate helpful and appropriate responses.

Our main contributions are as follows:

- We introduce REPO as a representation-based preference optimization method for detoxifying LLMs, providing a novel approach to removing toxic behaviors.
- We demonstrate that REPO achieves state-of-the-art performance, significantly reducing toxicity while preserving model utility, and showing superior robustness to a suite of adversarial attacks.
- We provide a detailed mechanistic analysis comparing REPO to both output-space and other representation-space methods. We show that representation-based objectives induce deeper changes in the network and that REPO 's modifications are uniquely localized to toxic tokens and the specific neurons that encode toxicity.
- Through targeted ablations, we identify the causal factors behind this behavior, showing that the representation-level objective is responsible for the depth of the edits, while the token-level granularity is critical for their surgical precision.

## 2 BACKGROUND

We begin by defining the preference optimization setting in terms of unlearning terminology. Each prompt $x_p$ is paired with two continuations: a *retain continuation $x_r$* (nontoxic) and a *forget contin-*

*uation* $x_f$ (toxic). The goal is to modify the model such that it erases information tied to the forget continuations while preserving its ability to generate fluent retain continuations.

Formally, we define a dataset of triples $\{(x_p, x_r, x_f)\}_{i=1}^N$. Each continuation is assigned a binary domain label $d \in \{0, 1\}$, where $d = 0$ for retain continuations and $d = 1$ for forget continuations. This framing allows us to cast preference optimization as a domain-adversarial problem: the model must maintain generation quality while ensuring that forget continuations cannot be distinguished from retain continuations in representation space.

Our method, which we present in Section 3, builds on the adversarial learning principle introduced in Domain-Adversarial Neural Networks (DANN) (Ganin et al., 2016). In DANN, the objective is to learn representations that are discriminative for the main classification task while being indiscriminate with respect to domain (source vs. target). This is achieved by training a domain regressor to separate domains, while the feature extractor is updated through a gradient reversal layer (GRL) so that domain membership becomes harder to detect.

SURE (Sepahvand et al., 2025) applies this idea to the problem of unlearning in classification models. Instead of distinguishing source from target domains, the regressor is trained to separate forget samples from a held-out validation set drawn from the same distribution. The feature extractor is updated adversarially, erasing parts of the representation that distinguish the forget set, while still preserving performance on the retain set. This enforces representation erasure, enabling selective unlearning with utility comparable to an oracle model retrained from scratch without the forget set.

Both DANN and SURE employ Representation Erasure (RE), i.e., an additional term on the objective that seeks to erase certain prescribed differences in representation. In DANN, RE is used to erase the difference in representation between source and target. In SURE, RE is used to erase the difference between forget and (a held out copy of) retained data.

## 3 INTRODUCING REPO FOR DETOXIFYING LLMS

In this work, we combine RE with preference optimization to detoxify large language models. In particular, we study a pretrained LLM, consisting of a stack of transformer blocks $G_f(\cdot; \theta_f)$, each mapping tokenized inputs (prompt + continuation) into hidden representations, that are then transformed by linear unembedding layers $G_y(\cdot; \theta_y)$ to produce next-token distributions. For detoxification, we attach the regressor $G_d(\cdot; \theta_d)$ via a gradient reversal layer (GRL) at the final transformer block, as illustrated in Figure 1. This regressor is trained to separate retain (nontoxic) from forget (toxic) continuations of the same prompt, while the feature extractor is updated adversarially to erase this distinction. This seeks to erase the difference in hidden representations of toxic and nontoxic continuations, while preserving the structure needed for fluent generation.

Training combines two coupled objectives. The first is the retain loss, which preserves the behavior of the original model on nontoxic continuations. We minimize the KL divergence between the output distributions of the unlearned model $G_y(\cdot; \theta)$ and the frozen reference model $G_y(\cdot; \theta^{\text{ref}})$ (the original model prior to unlearning) on retain samples:

$$\mathcal{L}_{\text{retain}} = \frac{1}{|D_r|} \sum_{(x_p, x_r) \in D_r} L_y\big(G_y(G_f(x_p, x_r; \theta_f); \theta_y),\ G_y(G_f(x_p, x_r; \theta_f^{\text{ref}}); \theta_y^{\text{ref}})\big), \qquad (1)$$

where $L_y$ denotes the KL divergence between the token-level predictive distributions of the two models.

The second objective is the domain-adversarial loss, which encourages the model to make retain and forget continuations indistinguishable in representation space:

$$\mathcal{L}_{\text{adv}} = \frac{1}{|D_r| + |D_f|} \sum_{(x_p, x) \in D_r \cup D_f} L_d\big(G_d(R(G_f(x_p, x; \theta_f)); \theta_d), \mathbf{1}[x \in D_f]\big), \qquad (2)$$

where $L_d$ is the binary cross-entropy loss, $R$ is the gradient reversal layer, and $\mathbf{1}[x \in D_f]$ is the domain label.

The overall training objective, $\mathcal{L} = \alpha\, \mathcal{L}_{\text{retain}} + (1 - \alpha)\, \mathcal{L}_{\text{adv}}$, interpolates between the retain and adversarial losses, with $\alpha \in [0, 1]$ controlling the trade-off between preserving the original model behavior and enforcing unlearning.

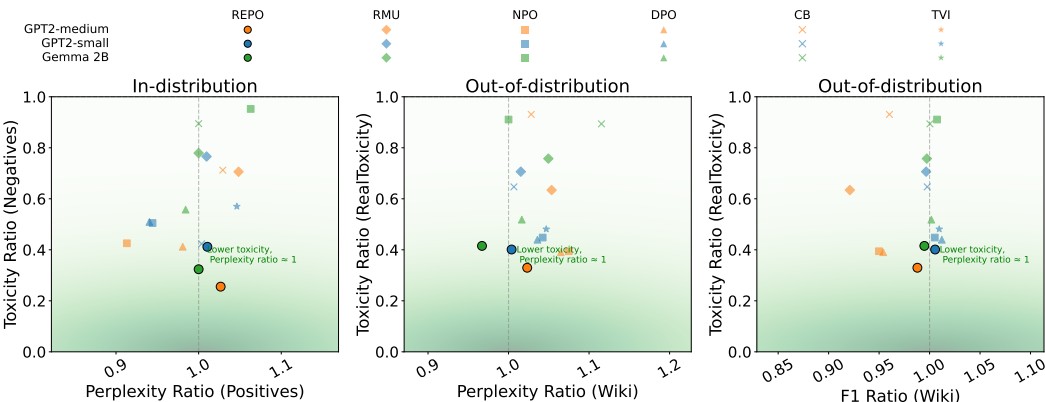

Figure 2: Detoxified models vs reference. **(Left)** Perplexity vs. toxicity ratios on PairToxicity (in-distribution); **(Middle)** Perplexity vs. toxicity ratios on WikiText/RealToxicity (out-of-distribution); **(Right)** $F_1$ ratio on WikiText vs. out-of-distribution toxicity. Each point is a model–method pair. The green gradient highlights lower toxicity and ratios near 1, darkest at the ideal point $(x = 1, \ y = 0)$. Dashed gray lines mark ratio = 1 for easy comparison to the reference.

## 3.1 EVALUATION METRICS

We evaluate our approach along two complementary dimensions: (i) its effectiveness in removing toxic behaviors while preserving general capabilities; this is often referred to as unlearning-utility trade-off, and (ii) its robustness against adaptive attacks aimed at reactivating toxic behaviors. Below we describe the metrics used in each case.

### 3.1.1 EFFECTIVENESS

**Toxicity Score.** Following prior work (Geva et al., 2022; Lee et al., 2024), we evaluate toxicity using the Perspective API[1], an automated tool for toxicity detection that estimates the probability a continuation would be perceived as toxic.

**Utility.** Utility is evaluated using perplexity and $F_1$ score on WikiText (Merity et al., 2017) , a neutral dataset excluded from unlearning. Perplexity, defined as the exponentiated average negative log-likelihood of the ground-truth continuation, measures how well a model predicts reference text. We report perplexity for both the unlearned and the reference model, i.e. the original model before unlearning, which is regarded as a high-utility reference point; differences between them provide a proxy for divergence from the distribution of the original pretrained model. $F_1$ is defined as the harmonic mean of precision and recall, where precision is the fraction of generated tokens appearing in the ground-truth continuation, and recall is the fraction appearing in the model's generation.

### 3.1.2 ROBUSTNESS

A key challenge in unlearning is robustness: even if a model forgets toxic behavior initially, adversaries may attempt to recover it. We consider three attack strategies studied in the unlearning literature (Wang, Zhang, et al., 2025; Łucki et al., 2025; Hu, Fu, et al., 2024): relearning, orthogonalization, and enhanced GCG. For the latter two, model weights remain frozen and only inference-time manipulations are applied, whereas relearning modifies the model via fine-tuning. Attack effectiveness is quantified by comparing the toxicity of generations from the unlearned model before and after the attack: an increase in toxicity indicates recovery of toxic behavior.

**Relearning Attack.** Prior studies have shown that fine-tuning can easily reverse alignment or unlearning, even when the fine-tuning data is small or consists of datasets with low mutual information with the forget set (Wang, Zhang, et al., 2025; Łucki et al., 2025; Hu, Fu, et al., 2024; Siddiqui et al., 2025). As in prior work (Łucki et al., 2025), we fine-tune unlearned models under two configurations: (i) on 10 examples from the forget set, and (ii) on 1000 examples from the retain set. The

---

[1]https://github.com/conversationai/perspectiveapi

former evaluates recovery under minimal direct exposure, while the latter tests recovery using data with low mutual information with the forgotten knowledge.

**Orthogonalization Attack.** Previous work demonstrated that safety refusals can often be attributed to a direction in activation space (Arditi et al., 2024). Łucki et al. (2025) extended this idea to the unlearning setting. Following their approach, we compute an *unlearned direction* for each transformer block as the difference in mean activations between the reference and unlearned models on the forget set (Łucki et al., 2025; Belrose, 2023). At inference time, this direction is projected out of the hidden representations, thereby removing the offset introduced by unlearning and restoring toxic capabilities.

**Enhanced GCG Attack.** GCG attacks have been reported ineffective against representation-based unlearning methods such as RMU (Li et al., 2024; Łucki et al., 2025). To increase their strength, we adopt an enhanced variant that specifically tar-

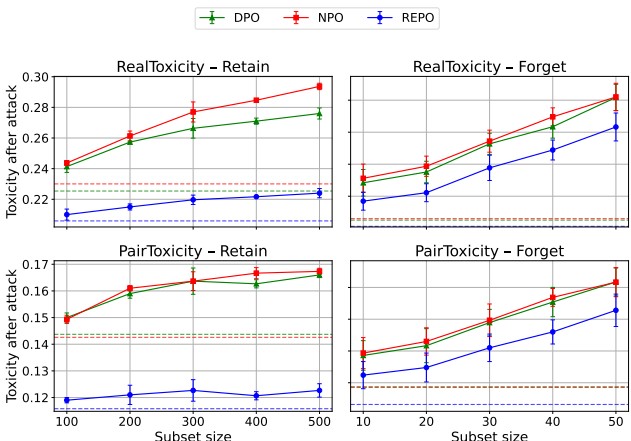

Figure 3: Average toxicity after the *Relearning Attack* for different subset sizes across methods on GPT2-small. **(Top)** Out-of-distribution toxicity (RealToxicity); **(Bottom)** In-distribution toxicity (pairwise set). Dashed horizontal lines indicate each method's baseline toxicity before the attack.

gets unlearning defenses (Łucki et al., 2025). Rather than minimizing the standard attacker loss toward generating a fixed affirmative target string (Zou, Wang, et al., 2023), the attack leverages the reference model as a malicious teacher. Concretely, adversarial prefixes are optimized with a distillation loss that aligns the unlearned model's hidden representations at selected layers with those of the reference model (Thompson and Sklar, 2024). This adaptation enables recovery of harmful behaviors that classic GCG cannot elicit.

## 4 EXPERIMENTAL DETAILS

**Data and Models.** Our evaluation relies on three datasets serving complementary purposes: a carefully crafted pairwise toxicity dataset for unlearning (Lee et al., 2024), WikiText-2 (Merity et al., 2017) for measuring generation quality, and RealToxicityPrompts (Gehman et al., 2020) for assessing out of distribution toxicity. We evaluate our approach on GPT-2 Small, GPT-2 Medium (Radford et al., 2019) and Gemma 2B (base) (Team et al., 2024). See Section D for further details.

**Baselines.** We compare REPO against two main families of alignment methods: *steering-based* and *fine-tuning–based*, the latter being further subdivided into *representation-* and *output-based*.

*Steering-based methods* act directly on hidden representations at inference time, modifying activations to suppress toxic behaviors without retraining, or even finetuning, the model. As a representative baseline, we adopt Toxic Vector Intervention (TVI) (Lee et al., 2024), which operates by subtracting identified toxic vectors from the model's activations during generation, providing a lightweight steering-style approach.

*Fine-tuning–based methods* explicitly retrain the model to remove undesired behaviors. We further divide them into two categories. *Output-space methods* operate directly on the model's output probabilities. Among them, we include preference-based objectives such as Direct Preference Optimization (DPO) and Negative Preference Optimization (NPO)(Wang, Zhang, et al., 2025; Łucki et al., 2025), which fine-tune the model to increase the likelihood of preferred continuations and decrease the relative likelihood of undesired ones. *Representation-space methods* operate on hidden activations. Examples include Representation Misdirection for Unlearning (RMU)(Huu-Tien et al., 2025; Kadhe et al., 2024) and Circuit Breakers (CB) (Zou, Phan, et al., 2024), which were

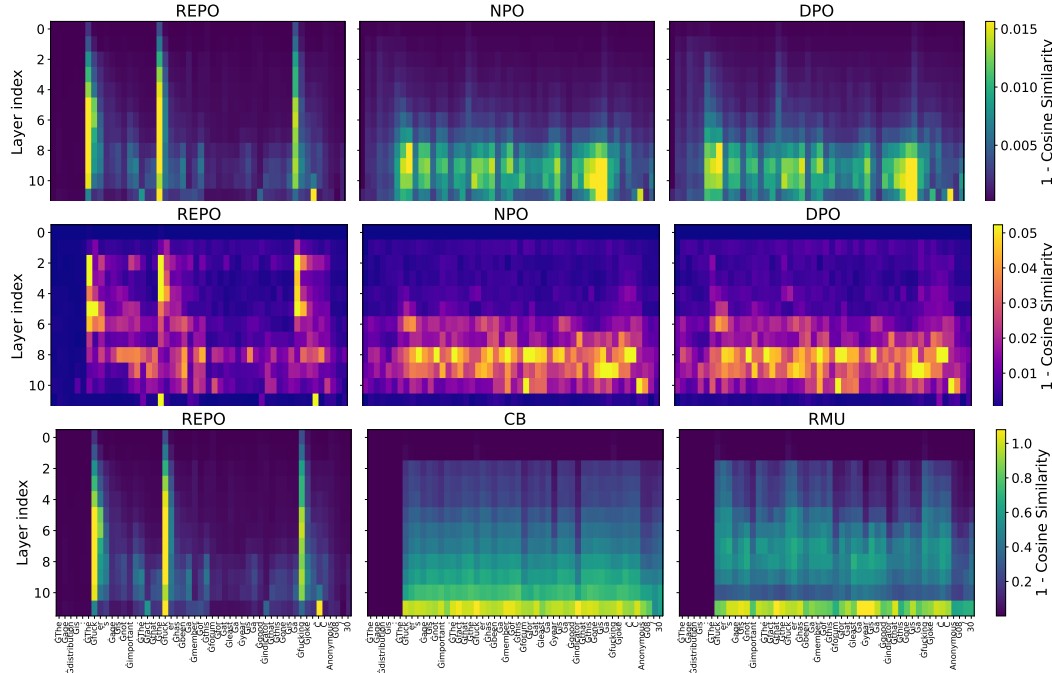

Figure 4: Layer–token distance heatmaps for different methods (columns) on a sample prompt. Columns left to right: REPO, NPO, DPO for the top two rows, and REPO, CB, RMU for the bottom row. **(Top)** $1-$cosine similarity between hidden states of the unlearned model and the reference model across GPT-2 small layers (vertical axis) and tokens (horizontal axis. Darker = more similar, yellow = larger difference. **(Middle)** $1-$cosine similarity between *attention submodule outputs* (before residual addition) of the unlearned model and the reference model across layers and tokens. **(Bottom)** Same as the top row, but for representation-based methods.

| | | REPO | NPO | DPO | RMU | CB |
|---|---|---|---|---|---|---|
| **GPT-2** | Relearning Forget (PairToxicity) | **.169(.116)** | .202(.143) | .200(.144) | .253(.215) | .438(.120) |
| | Relearning Retain (PairToxicity) | **.119(.116)** | .148(.143) | .148(.144) | .204(.215) | .124(.120) |
| | Relearning Forget (RealToxicity) | **.294(.206)** | .377(.230) | .357(.224) | .463(.363) | .678(.332) |
| | Relearning Retain (RealToxicity) | **.207(.206)** | .245(.230) | .237(.224) | .362(.363) | .314(.332) |
| | Enhanced-GCG (RealToxicity) | **.208(.206)** | .347(.230) | .660(.224) | .389(.363) | .393(.332) |
| | Orthogonalization (RealToxicity) | **.308(206)** | .335(.230) | .315(.224) | .525(.363) | .335(.332) |
| **Gemma-2B** | Relearning Forget (PairToxicity) | **.108(.083)** | .255(.247) | .169(.146) | .329(.206) | .161(.160) |
| | Relearning Retain (PairToxicity) | **.089(.083)** | .249(.247) | .169(.146) | .212(.206) | .162(.160) |
| | Relearning Forget (RealToxicity) | **.257(.215)** | .461(.439) | .304(.244) | .579(.356) | .402(.412) |
| | Relearning Retain (RealToxicity) | **.216(.215)** | .453(.439) | .304(.244) | .344(.356) | .421(.412) |
| | Enhanced-GCG (RealToxicity) | **.217(.215)** | .472(.439) | .269(.244) | 358(356) | .428(.412) |
| | Orthogonalization (RealToxicity) | **.217(.215)** | .442(.439) | .248(.244) | .357(.356) | .415(.412) |

Table 1: Robustness of unlearning methods on GPT-2 (Medium) and Gemma-2B across PairToxicity and RealToxicity. REPO dominates. Rows show evaluation setups; columns show methods. Each cell reports the toxicity score after the robustness attack, with parentheses showing the score immediately after unlearning but before any attack. Baseline toxicity scores (before unlearning): GPT-2 – 0.281 (PairToxicity), 0.513 (RealToxicity); Gemma-2B – 0.208 (PairToxicity), 0.486 (RealToxicity).

originally proposed to erase hazardous knowledge but here are adapted to the detoxification setting. While RMU maps toxic directions in representation space to random directions, CB severs identified causal pathways associated with harmful behavior.

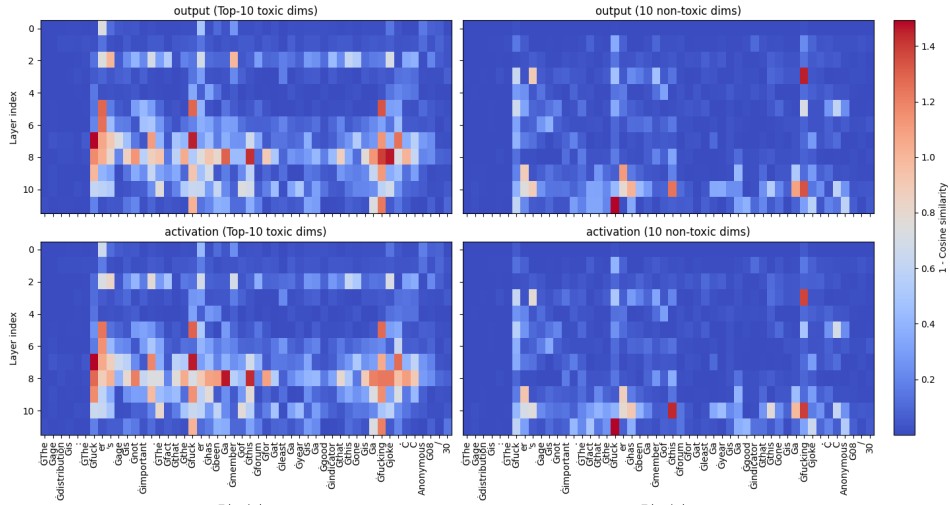

Figure 5: Layer–token residual-stream drift (1−cosine similarity) between the reference and REPO models for the same negative prompt. **(Top)** Differences in residual contributions (post-activation keys multiplied by value vectors); **(Bottom)** Differences in key activations. Within each row: **(Left)** Top-10 "toxic" dimensions (i.e., value vectors most aligned with the learned toxicity direction $W_{\text{toxic}}$); **(Right)** 10 non-toxic dimensions. Rows correspond to GPT-2 Small layers and columns to prompt tokens; darker colors indicate greater similarity and yellow larger drift.

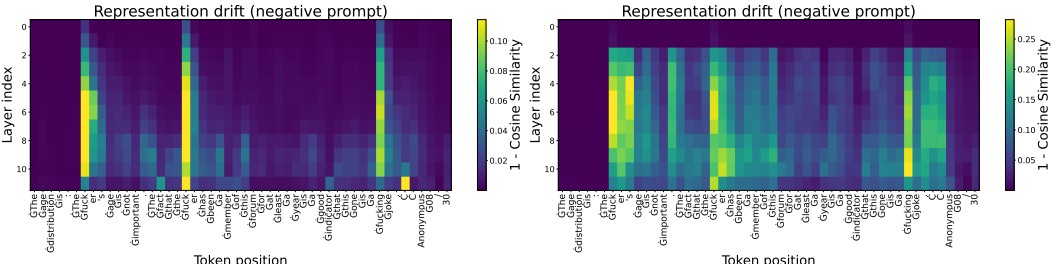

Figure 6: Layer–token representation drift (1−cosine similarity) for the same negative prompt under two discriminator input strategies in REPO: **Left** — individual tokens; **Right** — non-overlapping averaged segments. Darker colours indicate greater similarity, yellow larger drift.

## 5 PERFORMANCE EVALUATION

Our initial analysis focuses on several key performance aspects. As standard, we first examine the trade-offs between unlearning quality and model utility. This evaluation covers both in-distribution performance on the pairwise dataset used for unlearning, and out-of-distribution generalization on RealToxicityPrompts and wikitext-2. Here we see that REPO has superior performance across the board. We then evaluate robustness to various attacks, observing REPO's competitive performance.

**On the Trade-off of Mitigating Toxicity vs Preserving Utility.** Figure 2 reports results on the pairwise dataset. REPO achieves the lowest toxicity on negative (forget) samples, with a score of 0.0961, substantially outperforming NPO (0.1392), DPO (0.1506), and RMU (0.1527). Importantly, REPO also maintains comparable toxicity levels on positive (retain) samples, showing that the intervention does not degrade nontoxic generations. Perplexity results indicate that REPO increases the uncertainty of the model on toxic continuations (70.8 vs. 18.1 for the baseline), consistent with the goal of erasing toxic information, while leaving perplexity on retain samples largely unchanged relative to other methods. These findings suggest that REPO effectively targets toxic continuations without impairing general language modeling ability.

Figure 2 evaluates the methods on external benchmarks in order to assess out of distribution performance. On RealToxicityPrompts, REPO again yields the largest reduction in toxicity (0.2071 vs. 0.5062 for the baseline and 0.2374 for the next-best method, NPO). On wikitext-2, REPO achieves the best perplexity (23.6) and $F_1$ score (0.226), indicating that detoxification is not obtained at the cost of reduced fluency or predictive accuracy. Overall, across both in-distribution and out-of-distribution evaluations, REPO demonstrates more effective unlearning and stronger robustness than competing baselines.

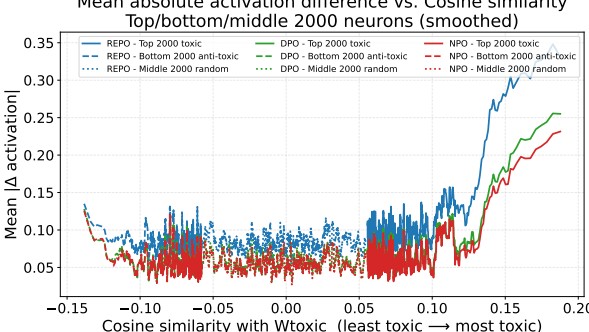

Figure 7: Mean absolute activation difference as a function of neuron toxicity alignment. Each curve shows the average absolute change in neuron activation between the unlearned model and the reference model, plotted against the neuron's cosine similarity to the learned toxicity direction $W_{\text{toxic}}$ (x-axis). (Solid lines) top 2 000 neurons aligned with $W_{\text{toxic}}$. (Dashed lines) bottom 2 000 (anti-aligned) neurons. (Dotted lines) 2 000 random remaining neurons. Colours indicate unlearning methods (REPO, DPO, NPO). Higher y-values indicate a larger mean absolute activation difference from the reference model; the plot is smoothed with a moving window of 20 for readability.

**Robustness to Attacks.** Table 1 examines robustness under adversarial attack settings. We evaluate three types of attacks: relearning, where the forget set is reintroduced through lightweight fine-tuning; enhanced GCG, which improves the success rate of adversarial prompting; and orthogonalization. REPO achieves the best overall robustness, consistently outperforming or matching the strongest baselines. In the relearning setting, REPO shows stronger robustness on retain samples, with toxicity at 0.207 compared to 0.270 for NPO and 0.289 for DPO. On forget samples, it achieves the second-best robustness, with toxicity remaining lower than most baselines after the attack and close to the strongest baseline RMU. Against enhanced-GCG, REPO achieves the lowest toxicity, at 0.252 compared to 0.295 for RMU and 0.379 for NPO. This demonstrates that REPO maintains robustness by resisting the recovery of toxic behaviors under this stronger adversarial attack.

## 6 THE EFFECTS ON REPRESENTATIONS AND WEIGHTS

The analysis in this section is focused on studying the mechanisms behind REPO's performance. We demonstrate that REPO has larger magnitude weight edits (Section E), but these edits result in more localized edits on conditional distributions of toxic words, and affect representations deeper in the network. Building on the analysis by Lee et al. (2024), we then inspect the changes in value and key vectors, observing that the biggest shift happens in dimensions most *and least* aligned with toxic directions. Our ablations reveal that these differences between REPO and other methods are due to two key algorithm design choices: (1) edits on the representations instead of output, resulting in bigger changes deeper in the network, and (2) REPO's optimization objective being at a token-level granularity – this ensures more localized shifts on the toxic word distribution.

**Changes in the intermediate states.** In Section E we show that REPO makes larger edits in weight space. Having observed that, we now examine how these changes affect the model's intermediate representations. Fig. 4 visualizes this by plotting the representational drift (1-cosine similarity) between the unlearned and reference models' hidden states across all layers for a sample toxic continuation. The heatmaps for REPO show that modifications are highly localized. Significant drift is concentrated in the network's deeper layers, and is confined almost exclusively to the columns corresponding to the toxic tokens, while the representations for adjacent tokens show minimal change. In stark contrast, DPO and NPO induce more diffuse, lower-magnitude changes that are spread across a broader set of tokens and layers. This analysis provides an intuition for REPO 's good utility-unlearning trade-off: it achieves effective unlearning by making targeted modifications to the representations of specific toxic inputs while preserving the integrity of non-toxic ones.

## 7 ABLATIONS OF ALGORITHMIC COMPONENTS

We conduct a series of ablations to dissect REPO's design and identify the sources of its effectiveness: representation-space edits, and token-level objective. We then provide evidence that REPO more aggressively targets the specific neurons most aligned with toxicity compared to baselines.

**Changing the token-level objective.**    To isolate REPO's components responsible for the localized edits, we conduct an ablation study on the granularity of REPO's adversarial objective. We compare our standard approach, where the discriminator evaluates each token's representation individually, with a variant where representations are averaged over non-overlapping segments before being passed to the discriminator. The results are visualized in Fig. 6. The left panel, showing the standard token-level objective, exhibits the highly localized representational drift previously discussed. In contrast, the right panel shows that using averaged segments causes this localization to vanish. The representational drift becomes diffuse, spreading across multiple tokens rather than being confined to specific ones. This diffusion in representation space correlates with a degradation in unlearning performance, yielding a worse utility-unlearning trade-off. This ablation provides strong evidence that the token-level granularity of REPO 's adversarial loss is a key mechanism responsible for the precision of its edits, which in turn contributes to its strong performance.

**The role of representation-based objective.**    Our analysis has shown that REPO's interventions are concentrated in deeper layers compared to output-space methods like DPO and NPO. To determine if this is a general property of representation-based unlearning, we now compare REPO with two other representation-based methods: Circuit Breakers (CB) and Representation Misdirection (RMU). The heatmaps in Fig. 4 (bottom row) confirm that this is indeed the case. All three representation-based methods predominantly alter the model in its later layers, suggesting that the depth of modification is a feature of targeting internal representations directly. However, the figure also reveals a critical distinction in the precision of these deep edits. While REPO 's changes are localized to specific toxic tokens, the interventions from CB and RMU are not. CB's edits appear to impact entire layers indiscriminately, and RMU's are scattered broadly across both tokens and layers. This comparison yields a key insight: while targeting representations helps focus unlearning on deeper parts of the network, it is REPO's specific token-level adversarial objective that provides the localization necessary for effective detoxification, a property that these other representation-based methods lack.

**Changes in neuron activations.**    Finally, we investigate how each method alters the activations of individual neurons based on their semantic roles. Following prior work, we first identify a toxic direction, $W_{\text{toxic}}$, using linear probing on the reference model's representations. We then measure the mean absolute change in neuron activations post-unlearning as a function of their value vectors' alignment with this direction. Fig. 7 reveals a consistent U-shaped pattern for all methods: the largest changes in activation occur in neurons that are most aligned or anti-aligned with W toxic, while neutrally-aligned neurons are minimally affected. However, the key distinction lies in the magnitude of this effect. For the neurons most aligned with the toxic direction, REPO induces a substantially larger change in activation compared to DPO and NPO. This finding suggests that REPO not only localizes edits to toxic tokens in the sequence but also concentrates its interventions on the very neurons most responsible for representing toxic concepts.

## 8 DISCUSSION

In this work, we introduced REPO, a novel method for detoxifying large language models by directly intervening on their internal representations. Our approach adapts the principles of domain-adversarial unlearning to force the representations of toxic generations to align with their benign counterparts. Experimental results demonstrate that REPO achieves a state-of-the-art reduction in toxicity while preserving model utility, and is highly robust to adversarial attacks, outperforming established baselines. A key finding from our mechanistic analysis is that REPO's representation-level objective induces deeper and more localized edits, focusing on toxic tokens and the specific neurons responsible for encoding toxicity. While highly effective for detoxification, the generalizability of REPO to unlearning other complex and undesirable behaviors, such as subtle social biases or factual inaccuracies, remains an important area for future investigation.

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

# Contents

## A  QUESTIONS WE ANTICIPATE

1. **Why did you choose models like GPT-2 and Gemma-2B base for evaluation?** Our choice was deliberate: these models are lightweight enough to support detailed *layer- and token-level mechanistic analysis*, which is central to the paper's contribution. Importantly, REPO is *model-agnostic* and scales naturally: the method only requires access to intermediate representations and a discriminator. Our experiments offer a compelling proof-of-concept with deep mechanistic evidence. Importantly, REPO's behavior is consistent across two distinct architectures (GPT-2, Gemma), suggesting architectural generality. Many unlearning methods (e.g., RMU, CB) were first validated on smaller scales before scaling up; we view our work as establishing the mechanistic foundation for future large-scale extensions.

2. **Why did you not test REPO on larger instruction-tuned models like Llama-2-7B or Mixtral?** Our experiments deliberately focus on smaller open models (GPT-2, Gemma-2B) to allow exhaustive mechanistic analysis (layer–token drift, neuron activation shifts, weight-space distances). These analyses would have not been feasible on 13B+ models due to cost and reproducibility barriers. Our goal is to provide a controlled, mechanistic demonstration. Scaling REPO is conceptually straightforward: it requires only a discriminator on hidden states. We are releasing code so the community can apply it to larger aligned models.

3. **Why are there no human evaluations or alternative detectors for toxicity?** We agree that multiple evaluators would enrich the results. For this submission, we prioritized comparability with prior ICLR/NeurIPS papers by using Perspective API, ensuring our baselines are on equal footing. Crucially, REPO does not optimize against Perspective, so it is detector-agnostic. Our mechanistic evidence (localized neuron edits, deeper layer shifts) shows that REPO changes the model itself, not just a metric. We view this as a stronger and more general guarantee than detector-specific scores.

4. **Why are ablations focused on token- vs segment-level?** We prioritized the ablation most central to REPO's novelty: token-level discrimination. Other knobs (loss weighting, discriminator depth) have standard effects and do not alter the mechanistic story. Our weight- and neuron-level analyses already show that REPO's behavior differs qualitatively from prior methods, and these structural differences (not hyperparameter sweeps) are what account for its robustness. Further ablations are left to future work due to space constraints.

5. **Does the use of synthetic toxic/non-toxic pairs introduce bias or limit generalization?** Synthetic pairs (via PPLM and greedy decoding) allow us to control for semantic similarity while isolating toxicity, which is essential for training a representation-level discriminator. This setup minimizes confounds such as topic or length, ensuring that REPO learns to erase toxic features rather than spurious correlations. Importantly, REPO's robustness evaluations (orthogonalization, relearning, GCG jailbreaks) demonstrate generalization to settings far outside the synthetic training distribution. In addition, REPO achieves strong performance on naturally occurring toxic continuations (RealToxicityPrompts), indicating that it transfers beyond synthetic contrasts.

6. **Are the baseline comparisons (to DPO, NPO, CB, and RMU) fair, and why not include RLHF-tuned models?** We implemented DPO and NPO using standard hyperparameters from their original papers, verifying that our implementations match reported performance. For representation-level baselines (e.g., CB, RMU), we reproduced them faithfully to ensure apples-to-apples comparison. We did not include RLHF-tuned models because REPO is not intended as a competitor to RLHF; rather, it is complementary. RLHF requires extensive preference data and large-scale tuning, while REPO can be applied post-hoc as a lightweight safety repair that directly edits hidden states. Thus, our focus is on representation-level methods, which are the most natural comparators—but REPO can also be layered on top of RLHF-trained systems.

7. **Is the enhanced GCG attack too unrealistic as a threat model?** We agree that access to the reference model is not always realistic, but we deliberately stress-tested REPO under *worst-case white-box assumptions*. The fact that REPO resists these extreme attacks strengthens confidence in its robustness to weaker, more realistic black-box jailbreaks. Our framing follows the *cryptographic principle of testing against the strongest adversary available*.

8. **Where can I find hyperparameters and training details?** See Section C.

9. **Why do the experiments focus only on toxicity, rather than other unlearning tasks?** We chose toxicity as a *representative and socially urgent case study*. The method, however, is general: REPO only requires a binary discriminator on hidden states. In principle, it can be applied to any capability removal (e.g., memorized data, unsafe skills). We see our toxicity experiments as a *first demonstration*, with generalization left for follow-up work.

## B   ON THE NOVELTY OF REPO AND HOW IT DIFFERS FROM SURE

The primary novelty of our proposed method, REPO, lies in its adaptation and extension of the representation erasure concept, originally developed for unlearning in classification models, to the domain of preference optimization for large language models. While it builds on the foundation of SURE, REPO introduces several key innovations in its framework, objective function, and mechanistic application that are specifically tailored for detoxifying generative models.

First, REPO reframes the unlearning problem as a preference optimization task. Unlike SURE, which aims to make a "forget set" of samples indistinguishable from a general held-out dataset, REPO leverages a pairwise data structure. Each prompt is associated with both a desired (non-toxic) and an undesired (toxic) continuation. This allows for a more targeted intervention: instead of matching a general distribution, the goal is to specifically align the representations of toxic outputs with their benign counterparts for the exact same context.

Another core REPO's design choice is its coupled training objective, which is designed to balance effective erasure with utility preservation. This objective combines two distinct components:

- A retain loss that explicitly preserves the model's behavior on non-toxic inputs. This is achieved by minimizing the KL divergence between the output distributions of the unlearned model and a frozen reference model on the retain samples. This component acts as a strong regularizer against degrading the model's general language capabilities.

- A domain-adversarial loss that drives the representation erasure. It uses a discriminator and a Gradient Reversal Layer (GRL) to adversarially train the model, making the hidden representations of toxic and non-toxic continuations indistinguishable.

Finally, a crucial mechanistic novelty in REPO is the token-level granularity of the adversarial objective. Our ablations reveal that applying the discriminator to individual token representations is responsible for the precision of REPO's edits. This design choice ensures that representational changes are highly localized to specific toxic tokens, preventing the diffuse, widespread modifications seen in other methods and contributing directly to REPO's strong performance and robustness.

## C   REPRODUCIBILITY STATEMENT

To facilitate reproducibility, we provide in Table 2 the exact hyper-parameters used for each method and model evaluated in this paper, together with their definitions. We also detail in Table 3 the

Table 2: Hyper-parameters used for each method and model. A dash (–) indicates the parameter is not applicable. For parameters listed as arrays in the configuration (e.g., two runs with $5 \times 10^{-6}$ for NPO on Gemma-2B), the table specifies this explicitly.

| Model | Method | Learning Rate (lr) | $\alpha$ | $\beta$ | $c$ |
|---|---|---|---|---|---|
| GPT-2-Small | REPO | $2 \times 10^{-6}$ | 0.2 | – | – |
| | DPO | $1 \times 10^{-6}$ | – | 0.5 | – |
| | NPO | $1 \times 10^{-6}$ | 0.2 | 0.5 | – |
| | RMU | $5 \times 10^{-6}$ | 0.95 | – | 500 |
| | CB | $1 \times 10^{-5}$ | 100.0 | – | – |
| GPT-2 Medium | REPO | $5 \times 10^{-6}$ | 0.2 | – | – |
| | DPO | $1 \times 10^{-6}$ | – | 0.5 | – |
| | NPO | $1 \times 10^{-6}$ | 0.4 | 0.5 | – |
| | RMU | $5 \times 10^{-6}$ | 0.95 | – | 500 |
| | CB | $5 \times 10^{-5}$ | 100.0 | – | – |
| Gemma-2B | REPO | $5 \times 10^{-5}$ | 0.5 | – | – |
| | DPO | $1 \times 10^{-5}$ | – | 0.2 | – |
| | NPO | $5 \times 10^{-6}$ | 0.8 | 0.5 | – |
| | RMU | $5 \times 10^{-5}$ | 0.95 | – | 500 |
| | CB | $1 \times 10^{-5}$ | 1000.0 | – | – |

training settings used across models, including the number of unlearning or relearning epochs, batch sizes, weight decay values, and other implementation choices. In addition, we describe the setup of our relearning attack experiments and the sampling procedures used for forget and retain sets. The full training and evaluation code will be released upon acceptance of the paper to enable independent verification and extension of our results.

**Hyper-parameter definitions.** Below we explain the roles of the hyper-parameters as used in our implementations (consistent with the original formulations when applicable):

- **lr:** learning rate used for parameter updates by the optimizer.
- **REPO — $\alpha$:** weight on the adversarial (discriminator) loss relative to the KL/reference-matching loss. It controls the trade-off between preserving similarity to the reference model and aligning the forget representations toward the retain representations in the shared space.
- **DPO — $\beta$:** scaling factor applied to the difference in log probabilities between the model and reference ($\Delta \log p$); it sharpens or flattens the preference logit before the log-sigmoid. Higher $\beta$ yields more aggressive preference gradients.
- **NPO — $\beta$:** scaling factor in the negative-preference term; $\alpha$ weights the forget loss relative to the standard LM loss on retain examples. Together they govern how strongly the model is pushed to forget and how much it is anchored to the retain examples.
- **RMU — $\alpha$:** interpolation weight between forgetting and retaining representations. The hyper-parameter $c$ defines the norm of the random "control" vector used to specify the forgetting direction against which the representation is aligned.
- **CB — $\alpha$:** coefficient on the circuit-breaker loss relative to the retain loss, determining how strongly the model is penalized when inner-product activations associated with forget features deviate from the desired retain alignment.

**Training and implementation details.** Beyond the hyper-parameters in Table 2, Table 3 summarises the key training settings we used across models and methods. These include the number of unlearning epochs, batch sizes, weight decay values, and learning rates used for the "relearning attack" experiments. All unlearning runs used a linear learning-rate warm-up of 100 steps. For DPO and NPO, we additionally clamped the logits to a fixed range (-30 to +30) to prevent numerical overflow and applied gradient-norm clipping to improve training stability.

Table 3: Training settings and implementation details for unlearning and relearning experiments.

| Model | Setting | Value | Notes |
|---|---|---|---|
| GPT-2 Small | Unlearning epochs | 10 | for all methods |
| | Batch size | 128 | for unlearning |
| | Weight decay | 0.001 | for all methods |
| | Relearning attack | wd $= 1 \times 10^{-5}$, lr $= 1 \times 10^{-5}$ | |
| | Gradient clipping | max_norm $= 10.0$ | DPO & NPO |
| GPT-2 Medium | Unlearning epochs | 10 | for all methods |
| | Batch size | 64 | for unlearning |
| | Weight decay | 0.01 | for all methods |
| | Relearning attack | wd $= 1 \times 10^{-5}$, lr $= 1 \times 10^{-5}$ | |
| | Gradient clipping | max_norm $= 10.0$ | DPO & NPO |
| Gemma-2B | Unlearning epochs | 5 | for all methods |
| | Batch size | 16 | for unlearning |
| | Weight decay | 0.01 | for all methods |
| | Relearning attack | wd $= 1 \times 10^{-4}$, lr $= 5 \times 10^{-5}$ | |
| | Gradient clipping | max_norm $= 10.0$ | DPO & NPO |

**Relearning attack.** For the relearning attack experiments, we fine-tuned the models for three epochs. We conducted two separate attack variants: (i) relearning on forget samples and (ii) relearning on retain samples. For the forget-based attack, we report the average over three independent runs, each using 10 randomly selected samples from the ToxicityPair dataset. For the retain-based attack, we likewise report the average over three runs using 100 randomly selected retain samples from the same dataset. In Figure 3 we show trends as we vary the set sizes; specifically, forget sizes {10, 20, 30, 40, 50} and retain sizes {100, 200, 300, 400, 500}. All reported values are averages over three independent runs.

## D    EXPERIMENTAL DETAILS

**Data.** The pairwise dataset, introduced in Lee et al. (2024), contains 24,576 prompt–continuation pairs constructed from sentences in Wikitext-2. For each prompt, we generate two continuations: a nontoxic continuation via greedy decoding, which forms the *retain set*, and a toxic continuation using PPLM (Dathathri et al., 2020) guided by a toxicity probe, which forms the *forget set*. This construction yields a *pairwise dataset* in which every prompt is associated with both a toxic and a nontoxic continuation, providing aligned examples for unlearning.

To measure preservation of generation capabilities, we use Wikitext-2, a standard language modeling benchmark consisting of Wikipedia articles, for evaluating perplexity and $F_1$. To measure toxicity reduction, we use the RealToxicityPrompts challenge set, which contains 1,199 prompts designed to elicit toxic outputs from language models.

**Models.** GPT-2 Medium is an autoregressive transformer trained on OpenAI's WebText corpus without any subsequent alignment or safety tuning. For Gemma 2B, we use the publicly available base checkpoints, which are pretrained models not fine-tuned for instruction following or safety; the aligned variants of these families (e.g., Gemma-Instruct) are deliberately excluded to ensure that detoxification is evaluated from raw pretrained models. For optimization, we apply full-parameter finetuning to GPT-2 (Small and Medium) given their smaller sizes, while for Gemma 2B we employ parameter-efficient LoRA finetuning.

## E    CHANGES IN THE WEIGHT SPACE

We examine the magnitude of modifications each unlearning method imparts on the model's parameters. Fig. 8 plots the average relative L2 distance between the weights of the unlearned and reference models at each Transformer block. A clear pattern emerges: REPO induces substan-

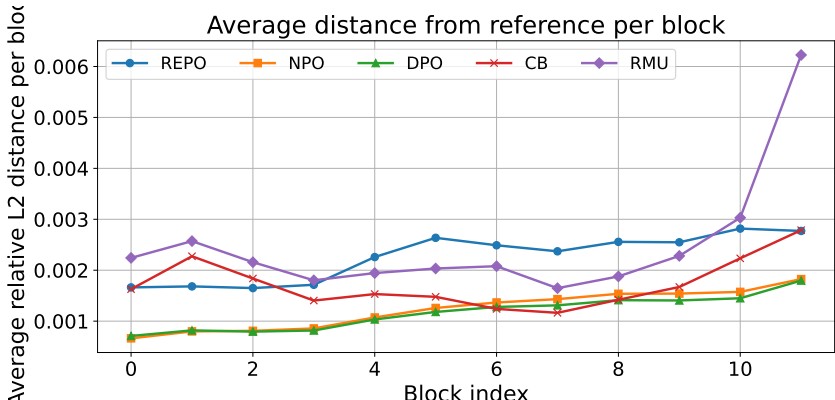

Figure 8: Average relative $\ell_2$ distance between unlearned models and the reference model at each Transformer block for REPO, NPO, and DPO.

tially larger weight-space edits compared to both DPO and NPO. While all methods tend to modify later layers more than earlier ones, REPO's updates are significantly greater, particularly from the middle to the final blocks of the network. Siddiqui et al. (2025) recently showed that unlearning algorithms that yield a larger L2 distance from the original model exhibit increased robustness to relearning attacks, which is consistent with our observation that REPO is significantly more robust against those attacks compared to DPO and NPO. For REPO, the larger weight-space edits are due to the method's design, which applies adversarial pressure directly to the hidden representations of the final transformer block. This architectural choice concentrates the learning signal in the deeper layers, compelling more significant parametric adjustments to align toxic and non-toxic representations. In contrast, DPO and NPO, which operate on output probabilities, distribute their updates more diffusely. While seemingly more disruptive, we will show in the following section that these larger weight-space modifications enable more precise, localized changes in the model's internal representations.

## F    Changes in Key and Value vectors

Plots in Figure 9 illustrate how each method affects the value and key vectors of the model across the top 2 000 neurons most aligned with the toxic vector $W_{\text{TOXIC}}$. Across all three methods (SURE, DPO, and NPO), the changes in both the value and key vectors are minimal, with cosine similarities between the pre- and post-unlearning weights remaining very close to one. For the most toxic neurons, our method induces a slightly larger reduction in cosine similarity, but this difference remains very subtle compared to the other two methods.

Despite the very subtle differences in key and value weight changes between our method and DPO/NPO, these small adjustments produce a markedly larger shift in the corresponding activations. Specifically, SURE yields a greater change in the key activations of those same neurons compared to DPO and NPO. In other words, even minor adjustments to the key and value weights, when guided by our adversarial alignment objective, are sufficient to shift the internal representations so that activations associated with toxic features are suppressed. This effect can be seen most clearly in the bottom row of Figure 9, where the mean absolute activation difference increases sharply for neurons most strongly aligned with toxicity. This demonstrates that SURE achieves detoxification primarily through targeted changes in the internal activations, rather than large weight updates, resulting in a more precise and controlled unlearning effect.

## G    Use of Large Language Models

Large language models were employed as an auxiliary tool during the preparation of this paper. In particular, they were used to (i) critique and suggest improvements to draft sections, (ii) assist in

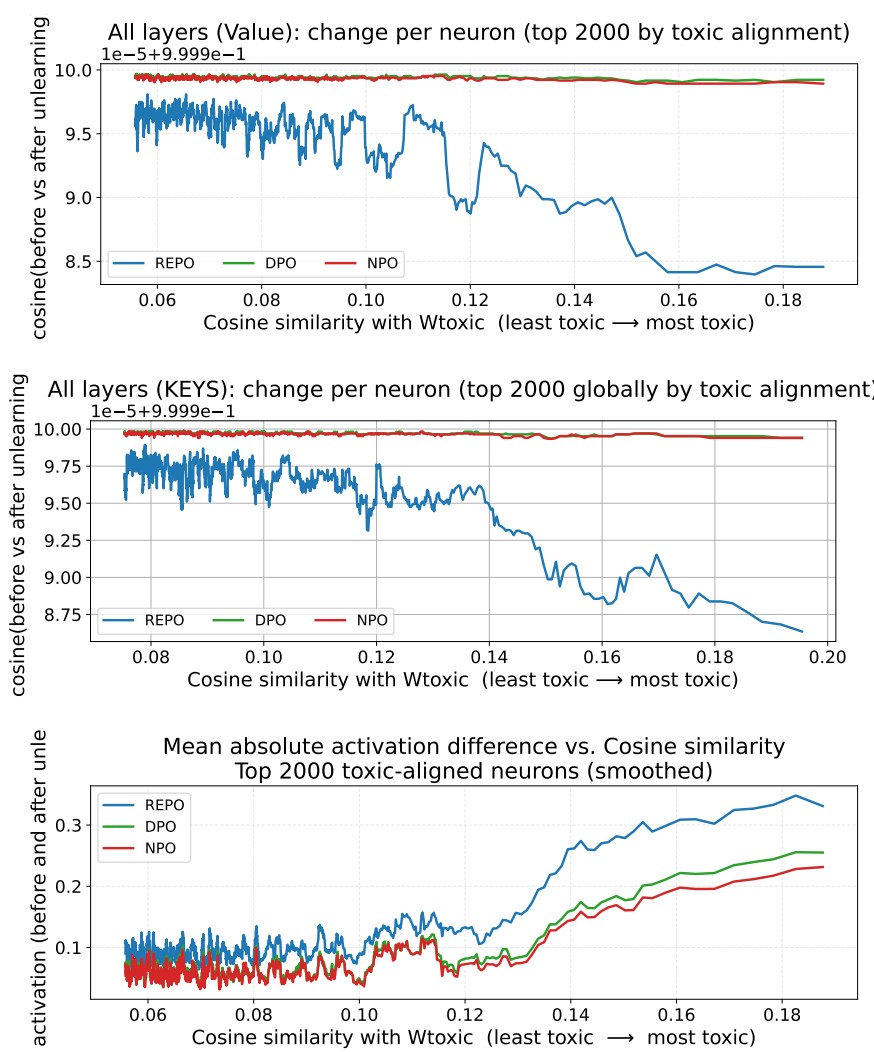

Figure 9: Comparison of how unlearning methods affect model internals. Top: cosine similarity between pre- and post-unlearning value vectors for the top 2000 toxic-aligned neurons. Middle: cosine similarity for key vectors of the top 2000 globally toxic-aligned neurons. Bottom: mean absolute activation difference vs. cosine similarity for the same neurons. Each curve shows REPO, DPO, and NPO behaviour as a function of cosine similarity with $W_{\text{toxic}}$ (left = least toxic, right = most toxic).

polishing the language of drafts or selected passages, and (iii) offer tips and guidance on how to improve the clarity and appearance of figures and plots.

