# OpenReview forum: "REPO: Detoxifying LLMs via Representation Erasure-based Preference Optimization"
_ICLR.cc/2026/Conference — Submitted to ICLR 2026_

### Official Review · Reviewer_w6HQ · 2025-10-30

**Soundness:** 2
**Presentation:** 2
**Contribution:** 1
**Rating:** 2
**Confidence:** 3

**Summary:**

This paper introduces REPO, an unlearning method aimed at detoxifying LLMs by intervening directly on their internal hidden representations. REPO adapts the principle of SURE (Sepahvand et al., 2025) to the preference optimization setting. Its core strategy is to preserve the representations of benign (nontoxic) generations while forcing the representations of toxic (forget) generations to converge toward their benign counterparts.

**Strengths:**

REPO demonstrates the effectiveness in detoxification while preserving general language capabilities. It achieves the lowest toxicity score on in-distribution and out-of-distirbution evaluations, substantially outperforming steering-based methods and other fine-tune based methods.

Prior output-level alignment techniques (like DPO and NPO) are vulnerable to adversarial prompting and relearning attacks, as linear probing suggests that harmful “directions” remain present in the model's representations. REPO achieves strong robustness against several advanced adversarial attacks, such as relearning, orthogonalization, and GCG attacks. The robustness is also evidenced by the representational drift  between the unlearned and reference models’ hidden states.

**Weaknesses:**

As the authors acknowledge in the Introduction, and Appendix B, REPO appears highly similar to SURE (and DANN), thus suggesting a lack of significant novelty. Though the authors say that they adopted SELU for preference optimization, the core components, specifically the retain loss and the adversarial loss, are directly adopted from SURE (and DANN), and therefore do not constitute a unique methodological innovation developed specifically for preference optimization.

The most significant difference is that SURE calculates the domain loss using the forget set and a validation set (non-trained set), whereas REPO calculates the domain loss using the toxic continuation and the non-toxic continuation. I don't understand why this achieves detoxification. In the case of SURE, I understand that unlearning is achieved because by bringing the hidden states of the forget set inputs closer to the hidden states of the non-trained set inputs, the model behaves as if the forget set inputs were never learned. On the other hand, for REPO, I don't understand why toxic text is unlearned by making the hidden states of the toxic continuation similar to the hidden states of the non-toxic continuation, and I question whether the adversarial loss is truly effective for the case of REPO.

As noted above, given the uncertainty regarding the effectiveness of the adversarial loss, I would like to see the results using only the retain loss. However, such an analysis is not conducted in the ablation study. Instead, the analysis focuses merely on several variants of the adversarial loss and the comparison against prior methods, which are not an ablation study.

**Questions:**

As mentioned in weakness, could you explain why the adversarial loss using the toxic continuation and the non-toxic continuation achieves unlearning (or preference optimization) step by step? Also, I would like to see the results using only the retain loss.

---

> ### Author Response · Authors · 2025-12-03
> **Internally inconsistent review**
>
> **Context and strategy of this rebuttal**
>
> Reviewer w6HQ’s concedes that REPO "achieves the lowest toxicity score [...], substantially outperforming steering-based methods and other fine-tune based methods" and "REPO achieves strong robustness against several advanced adversarial attacks, such as relearning, orthogonalization, and GCG attacks". In short, it's SOTA and more robust than existing SOTA methods like DPO/NPO. Strong accept? No...
>
> The reviewer rejects our work on the basis of two weaknesses: namely, REPO "lacks significant novelty" and, *at the same time*, "it is unclear why it works"(!). Just in case it is not already obvious, we will demonstrate that this argument is incoherent on the basis of being self contradictory.
>
> **Brief diatribe**
>
> Note: no reviewer is claiming that the application of a domain-adversarial method to detoxification (or preference optimization) is not novel, or that the empirical results are not impressive. And so, on the basis of a novel application and strong results on a very important problem of interest to academics and industry alike, we believe the paper is an obvious strong accept. We believe the AC should simply reject the premise of this review and move on, but we will attempt to decimate the review just in case.
>
> **The internal inconsistency**
>
> Recall that SURE is domain-adversarial training applied to unlearning in classification. The two losses are: a cross-entropy (CE)-based retain loss (keeping the model for forgetting about the data we are retaining) and the forget ("domain") loss (representing a discrimination task, predicting whether the final representation layer is that of a data point being retained or forgotten). Together, the losses forms a min-max objective: a discriminator attempts to disambiguate forget from retain, while the network attempts to fool the discriminator, while not straying too far from its current predictions.
>
> REPO takes this idea of domain adversarial training for unlearning, but to preference optimization and in particular detoxification. The retain loss is *replaced* by the KL divergence on prompts and (all subsets of) nontoxic continuations, keeping the model from moving too far away. The adversarial ("domain") loss represents the same discrimination task, but now the task boils down to detecting whether the final *representation layer* was produced by a context window with or without toxicity.
>
> The review asserts that REPO "is highly similar to SURE [...] thus lacks significant novelty." In one breath, they write "The most significant difference is that SURE calculates the domain loss using the forget set and a validation set (non-trained set), whereas REPO calculates the domain loss using the toxic continuation and the non-toxic continuation". In the *very next sentence* they write "I don't understand why this achieves detoxification." The reviewer wants to have their cake ("lack of significant novelty") and eat it too ("why do these insignificant changes work?!"). Maybe they aren't so insignificant?
>
> Note that the reviewer claims "In the case of SURE, I understand [why it works]". They then spin some wild theories: that the retain loss in REPO on its own is carrying the weight, and "I question whether the adversarial loss is truly effective for the case of REPO". Their big recommendation: "As noted above, given the uncertainty regarding the effectiveness of the adversarial loss, I would like to see the results using only the retain loss." This idea is **extremely confusing** (details in next comment), as the retain loss on its own is zero with zero gradient if you remove the adversarial loss! This would be a nonsense ablation. However, there's another interpretation: the reviewer has missed that REPO has a KL retain loss, and not SURE's CE retain loss.
>
> So, in fact, it seems the reviewer has missed that *the novelty of REPO lies in demonstrating that aligning negative representations to their positive counterparts (using paired continuations) effectively reduces toxicity, which we validate through extensive experiments.*
>
> Finally, the reviewer concludes with a mischaracterization of our ablations ("the analysis focuses merely on several variants of the adversarial loss and the comparison against prior methods, which are not an ablation study."). We can only explain this summary by assuming they did not read our paper very carefully, and, in particular, the entire appendix of ablations. Regardless, we add one more ablation (in the next official comment), showing that both of REPO's retain and adversarial loss are essential, and that SURE's CE retain loss is far inferior.
>
> We provide more detail in the next comment, as we've run out of space.

---

> ### Author Response · Authors · 2025-12-03
> **Some more detail.**
>
> **The adversarial loss is necessary; the retain loss cannot alone account for REPO's gains**
>
> The suggestion to test "retain-only" as if it were an informative ablation is very confusing. Our retain loss is the KL between the current and the original model on safe tokens. At initialization the two distributions are identical, hence the KL (and its gradient) is exactly zero. With alpha = 1 (retain-only), the model does not move. More fundamentally, the retain term never sees the toxic branch and has no pathway to suppress toxicity; the only force that acts on toxic representations is the adversarial ("domain") loss.
>
> Another possibility is that the reader has missed the fact that SURE and REPO do *not* share the same retain loss. (They share the same adversarial loss in a sense, even if the composition of the retain/forget sets is specialized different in the classification and detoxification settings). As stated above, REPO employs a KL divergence retain loss at the logit level, whereas SURE employs cross-entropy. To demonstrate the impact of this modification on detoxification performance, we conducted additional ablations on GPT2-small, comparing:
>
> * The reference model (no preference optimization/detoxification). **Note: This is also equivalent to REPO without the adversarial loss.**
> * Cross-entropy (CE) retain loss only (SURE objective with α = 0)
> * The original SURE objective (CE retain + adversarial)
> * REPO (KL retain + adversarial)
>
> Here is the table:
>
> | Method | Toxicity (Pos) | Toxicity (Neg) | RealToxicity | Wiki (perplexity) | Wiki (F1) |
> |---|---:|---:|---:|---:|---:|
> | Ref| 0.0460 | 0.2824 | 0.5123 | 28.0379 | 0.1930 |
> | CE retain-only | 0.0437 | 0.2367 | 0.4454 | 45.7689 | 0.1859 |
> | SURE | 0.0418 | 0.2021 | 0.3750 | 34.0385 | 0.1869 |
> |REPO|0.0446|0.1020|0.1913|28.2314|0.1930
>
> The results show that CE on the retain set only yields only minor toxicity reduction and severely harms perplexity (28 v. 34).  SURE (CE retain + adversarial loss) improves toxicity more, but underperforms REPO in terms of perplexity (28 v. 34). REPO, by comparison, achieves far stronger toxicity reduction (0.5123 to 0.1913) while maching the reference models perplexity.
>
> **REPO delivers the best of both worlds; alternatives are not comparable**
>
> As more evidence of novelty: REPO can be viewed as an improvement on RMU (random remapping) and CB (orthogonalization), two unlearning techinques that operate by modifying representations, which have been applied to model editing.  Our new experiments on RMU and CB reveal that they are blunt geometric interventions that shatter local semantics; when a toxic token appears, we show that subsequent generation degrades immediately. DPO/NPO alter outputs but do not remove the internal representational affordances that enable toxic generation, so the edits are superficial and easy to undo. REPO's token-level invariance removes those affordances while the KL retain keeps capability anchored, delivering detoxification that is both stronger and more durable.
>
> **Bottom line for the AC**
>
> The review's three pillars---"REPO is just SURE", "SURE, I understand", and "we don't understand why REPO works"---cannot all stand. The reviewer's suggest ablations are either nonsensical or miss a key difference between REPO and SURE in REPO's KL retain loss. REPO's design differs in exactly the ways that matter for preference-trained generative LMs. In conclusion, REPO's mechanism is empirically validated, and its empirical record shows the strongest detoxification and robustness without the utility collapse seen in prior representation-space methods. The correct outcome is to publish this work.

---

### Official Review · Reviewer_Hgpg · 2025-11-01

**Soundness:** 3
**Presentation:** 2
**Contribution:** 2
**Rating:** 6
**Confidence:** 2

**Summary:**

This paper proposed REPO, a novel method for detoxifying large language models by removing toxicity at the representation level during unlearning. Building on the SURE method, REPO achieves detoxification by aligning the LLM's internal representations of non-toxic and toxic continuations of the same prompt, so they appear indistinguishable to a toxicity classifier. Meanwhile, the toxicity classifier is trained adversarially to further distinguish the aligned representations produced by the LLM. The authors show that the GPT-2 and Gemma-2B models unlearned using REPO have lower toxicity compared to fine-tuning based and steering based detoxification. Moreover, the REPO-trained models are also more robust to a variety of relearning-attacks, GCG, and orthogonalization.

**Strengths:**

1. The paper is well written and easy to follow. While representation erasure–based unlearning has already been used by [1] for a classification task, this paper effectively extends it to text generation by adding a constraint that preserves the model’s generative utility.
2. The performance of REPO is generally robust. Compared to fine-tuning based and representation steering-based detoxification, models trained using REPO have lower toxicity both before and after variety of attacks.
3. The detailed analysis of REPO's effects on toxicity representation at layer (Figure 4, 5, 6) and neuron level (Figure 9) are convincing.

References:
[1] Sepahvand, Nazanin Mohammadi, et al. "Selective unlearning via representation erasure using domain adversarial training." The Thirteenth International Conference on Learning Representations. 2025.

**Weaknesses:**

1. Only tested on two small models (GPT-2 and Gemma-2B). Although small and lightweight models allow the authors to conduct more detailed mechanistic analysis, knowing whether the performance of REPO scales to larger and more recent LLMs is still important for validating the method's practical value.
2. REPO outperforms the fine-tuning based and steering based detoxification, but the adversarial training of its discriminator module also introduce additional computational overhead, which should be discussed.
3. The paper evaluates the utility of unlearned models by their perplexity on the WikiText dataset. To support the claim that REPO preserves general utility, performance on benchmarks such as HellaSwag, WinoGrande, OpenBookQA, or MMLU would be more informative.
4. The parameterization of the discriminator is underexplored. Using a linear regressor implicitly assumes toxicity is linearly represented, yet prior work [2] suggests toxicity may reside in a multi-dimensional subspace. It would be valuable to test whether a non-linear discriminator improves performance.
5. (Minor) The prompts along the x-axis in Figures 4–6 are difficult to read. Including the full prompts in the captions and highlighting toxic tokens in the heatmaps would improve clarity. Also, consider removing the extraneous “$\dot{G}$” characters from tokens.
6. (Minor) Figure 1 should be moved closer to Section 3 for better readability of the method description ($G_d, G_y$, etc.)

Reference:
[2] Uppaal, Rheeya, et al. "Model editing as a robust and denoised variant of dpo: A case study on toxicity." arXiv preprint arXiv:2405.13967 (2024).

**Questions:**

1. Have the authors tested REPO on removing representations of other undesirable features, such as bias and unsafe knowledge, which are more multi-faceted compared to toxicity? Does the performance of REPO generalize?

---

> ### Author Response · Authors · 2025-12-03
> **Should be more confident**
>
> We thank the reviewer for their careful reading of our paper. We address the technical concerns (discriminator complexity) below. To the AC: We believe our response would have permitted the reviewer to update their confidence, as they clearly understood what was going on.
>
> **Evaluation on general benchmarks (Weakness 1):** We agree that evaluating REPO on a downstream benchmarks would strengthen the paper. To address this request, we conducted an evaluation using the MMLU benchmark on the Gemma-2b model. (Recall that MMLU is multiple choice, with four choices.) We excluded the other two smaller models as their original, unlearned accuracy scores were not significantly higher than random. We tested all five unlearned model variants (REPO, DPO, NPO, RMU, and CB), as well as the original reference (Ref) model, on the MMLU benchmark:
>
> | Method | Ref       | REPO      |DPO       | NPO       | CB        | RMU |
> |--------|-----------|-----------|-----------|-----------|-----------|-----------|
> | Accuracy (mean ± std) | 0.418 ± 0.095 | 0.422 ± 0.091 | 0.422 ± 0.098 | 0.423 ± 0.096 | 0.418 ± 0.095 | 0.411 ± 0.093 |
>
> The average accuracy across methods was very similar, ranging from 0.411 to 0.423. We also compared accuracy for individual subjects and found that differences across methods were small. Any changes in the predicted probabilities were too minor to significantly affect the classification decisions that determine accuracy on MMLU.
>
>
> **Computational overhead of the discriminator (Weakness 2):** The overhead is negligible.  The discriminator is a small two-layer fully connected network mapping with dimensions $D_{\text{model}} \to 16 \to 2$. Compared to the full forward and backward passes through the model, this extra computation is effectively zero. The training time is dominated entirely by the LLM's gradient computation, not the discriminator.
>
> **Discriminator parameterization (Weakness 4):** In REPO, unlike SURE, we use a two-layer fully connected discriminator. This was not clearly mentioned in the paper and will be added in the final version. To address the reviewer’s concern, we compared a one-layer versus a two-layer discriminator on GPT‑2 small:
>
> | Method             | Toxicity (Pos) | Toxicity (Neg) | RealToxicity | Wiki (perplexity) | Wiki (F1) |
> |--------------------|----------------:|----------------:|-------------:|---------:|--------:|
> | Reference     | 0.0461          | 0.2812          | 0.5120       | 28.0379  | 0.1930  |
> | One-layer Disc.    | 0.0451          | 0.1356          | 0.2396       | 28.2567  | 0.1952  |
> | Two-layer Disc.    | 0.0446          | 0.1020          | 0.1913       | 28.2314  | 0.1937  |
>
> These results show that the two-layer discriminator is more effective than the one-layer version at reducing toxicity for both in-distribution (Negative Samples) and out-of-distribution data (RealToxicity), while perplexity, F1 score, and toxicity on positive samples remain essentially unchanged.
>
> **Figure readability, minor (Weakness 5-6):** We will include the full prompts in the captions of Figures 4 through 6, highlight toxic tokens in the heatmaps, and remove extraneous characters. We also agree that Figure 1 should be moved closer to Section 3 for better readability.
>
> **Generalization to other undesirable features (Question 1):** We have not tested REPO on other undesirable features such as bias or unsafe knowledge, primarily due to dataset limitations. Methods like REPO and DPO rely on pairwise datasets, where each prompt has both a desirable and an undesirable response. To our knowledge, such datasets do not currently exist for other applications, and would potentially require synthetic data generation - an interesting research question that’s beyond the scope.

---

### Official Review · Reviewer_b5qV · 2025-11-06

**Soundness:** 2
**Presentation:** 1
**Contribution:** 2
**Rating:** 2
**Confidence:** 4

**Summary:**

The paper addresses the detoxification of large language models (LLMs) through the lens of machine unlearning. The proposed approach aims to make toxic and non-toxic representations indistinguishable by applying an adversarial loss at a particular layer of the model. The method uses two components:

- $L_{retain}$: to preserve model performance on non-toxic data.
- $L_{adv}$: to enforce indistinguishability between toxic and non-toxic representations.

The idea is positioned as a generalization of unlearning methods to the alignment / detoxification setting.

**Strengths:**

- Provides an interesting attempt to connect recent machine unlearning approaches with alignment objectives, offering a conceptual link between the two, though not entirely novel.
- Addresses a timely and practically important goal "mitigating toxic behavior in LLMs".

**Weaknesses:**

1. The central claim "making toxic and non-toxic representations indistinguishable" requires more motivations. It is unclear why this would correspond to unlearning or guarantee behavioral safety.

2. The adversarial formulation is confusing. Minimizing the binary cross-entropy loss $L_{adv}$  appears to promote discrimination rather than indistinguishability. The paper should explicitly clarify that $G_d$​ (the discriminator) is trained to maximize $L_{adv}$, while $G_f$ (the forget model) is trained to minimize it, following a standard minimax setup.

3. There is no formal definition of the datasets $D_r, D_f$.

4. The role of $L_{retain}$ is vague: it is unclear whether it preserves token-wise predictions (during gernation of $x_r$) or sequence-level consistency (next token generation after $x_p, x_r$). If it compares only next token generation after $x_p, x_r$, the objective may not preserve semantics effectively.

5. It seems the idea is applied to the block M (typically the last lyer). However, the model modification (training) scope (only last layer vs. full model) is not clearly specified, yet this choice crucially affects both unlearning strength and computational cost.

6. Section 2 is redundant and reads like an extended introduction; this space could instead clarify mathematical definitions and literature survey.

7. The paper insufficiently engages with prior work on alignment methods such as DPO or PPO. Also, it would be better to describe the key ideo of unlearning method SURE. Without situating the method among these, its novelty and relevance to the alignment field remain unclear.

**Questions:**

see weaknesses

---

> ### Author Response · Authors · 2025-12-03
> **Confidently uncertain**
>
> This review argues for rejection, confidently. That confidence is, however, inconsistent with the reviewer's list of weaknesses, which betrays a considerable amount of uncertainty about key aspects of the paper. **How can a reviewer who is confused about so many basic aspects be so confident in their assessment?**
>
> The reviewer lists 7 weaknesses. We believe the most serious are 1, 4, 7.
>
> Briefly, **Weaknesses 2, 3, 5, and 6 are trivial to address,** as they relate to explaining aspects of past work or trivial missing definitions (which are somewhat obvious from context, since an LLM can infer them when asked). We're not really sure whether the reviewer is actually confused or they are sympathizing with future readers and ensuring we make changes for clarity. Regardless, we will clarify these in detail. Due to space, we will post these clarifications in a subsequent Official Comment.
>
> **Weaknesses 7 is very puzzling**. It reads "The paper insufficiently engages with prior work on alignment methods such as DPO or PPO. Also, it would be better to describe the key idea of unlearning method SURE. Without situating the method among these, its novelty and relevance to the alignment field remain unclear."
>
> *Regarding our not comparing with DPO and PPO*: we engage **heavily** with DPO and NPO. Figures 2, 3, 4, 5, 8, and 9 all directly compare our method REPO against DPO and NPO (a more recent variant of DPO). All of our experiments are towards teasing apart REPO and DPO or REPO and other representation-based approaches.
>
> *Regarding not explaining SURE adequately*: The reviewer argues to drop Section 2 (which explains SURE in part) and we suspect they missed Appendix B, where we provide even more background on SURE and its relationship with REPO.
>
> *Regarding novelty and alignment*, we think we've presented ample evidence that REPO is far superior to DPO in terms of robustness and far superior in utility to other representation-based unlearning methods, when applied to detoxification in the same way they've been applied to model editing.
>
> **Finally, Weaknesses 1 and 4 relate to a single typo.** If we had had the opportunity to clarify this typo, we believe this reviewer  would have seen the paper under new light. If we had to guess, they'd probably have given us a 10, or something close.
>
> The typo is a missing summation in both Eq 1. (retain loss $L_{\text{retain}}$), and Eq 2. (the adversarial loss $L_{\text{adv}}$). In particular, both are missing a $\sum_t$ over all causal masks, i.e., all prefixes of tokens 1:t . In other words, we are doing the standard thing of training on all causal masks. Without this summation, the KL divergence in $L_{\text{retain}}$ would only preserve the distribution for the token generated *after* the last token of the nontoxic continuation. That's a rather strange goal, which the reviewer notices in Weakness 4 and basically says "that's strange, why would that achieve the goal and why didn't you do the standard next-token thing". Indeed, with the summation, the $L_{\text{retain}}$ loss does the standard, sensible thing: it preserves the next-token distribution at every token in the nontoxic continuation. The $L_{\text{adv}}$ loss also does the sensible thing: it serves to push the representation for every causal mask of a toxic continuation to be more like the representation for every causal mask of a nontoxic continuation.
>
> With this typo explained, let's look at these two weaknesses.
>
> **Weakness 4**: This weakness is literally the reviewer asking about this missing summation in the $L_{\text{retain}}$ loss. We agree with the reviewer that the terms don't really make much sense without the $\sum_t$ and that they wouldn't really serve to keep the model from shifting too much.
>
> **Weakness 1**: This weakness is really the corollary of Weakness 4 applied to the adversarial loss, $L_{\text{adv}}$: the reviewer doesn't understand why REPO should work, and it would absolutely not work if we were only trying to make the toxic and nontoxic representations for the final token indistinguishable. Instead, with both losses defined appropriately, the retain loss keeps the model faithful to its nontoxic part, while the adversarial loss serves to erase the model's representation (features) of toxicity in its context, at any point in generation. Once it loses the ability to represent toxicity, it cannot generate it.
>
> .
>
> **In closing**, we feel this review is a rather negative view on our work. It does not seem to acknowledge the considerable empirical evidence we have amassed that REPO is a very effective and robust approach to detoxification. The application of domain adversarial objectives to preference optimization deserves further study, and so we urge you to reconsider this review as if we had had the chance to address this reviewer's considerable uncertainty.

---

> > ### Author Response · Authors · 2025-12-03
> > **Further details**
> >
> > Below we provide more detail on how we address each Weakness.
> >
> > **We make extensive comparison with DPO & NPO (Weakness 7):**
> > We note that the paper relies heavily on comparisons to DPO and NPO to demonstrate the advantages of representation-based methods over output-based methods. Figures 2, 3, 4, 5, 8, and 9 all directly compare REPO against DPO and NPO.
> >
> > **Our Background section covers SURE, which you ask for (Weakness 6 but also 7 and 2):** Section 2 (Background) is intentionally structured to provide the necessary foundation for Section 3, by introducing REPO by way of explaining DANN/SURE in detail, and situating our method relative to prior unlearning approaches. Appendix  A also clarifies why RLHF was not included as a baseline. Finally, Appendix B provides yet more context for SURE and how it relates to REPO. Nevertheless, we will see if there are any aspects of SURE we do not cover adequately, while we will look to streamline Section 2.
> >
> > **Adversarial formulation clarification (Weakness 2):**  The gradient reversal layer (GRL) implements the min-max optimization of $L_{adv}$, as established in the domain adaptation literature (DANN, Ganin et al.’16). Parameters above the GRL ($\theta_d$, discriminator) are optimized to improve discrimination, while parameters below ($\theta_f$, feature extractor) are optimized to fool the discriminator. As in DANN (Eq. 2), the GRL multiplies the gradient by $−1$ on the path to $\theta_f$, allowing all parameters to be optimized simultaneously using standard minimization.
> >
> > **Clarification on the retain objective (Weakness 4):** $L_{retain}$ is the KL divergence between the token-level predictive distributions of the unlearned and original model. The KL loss is summed over all tokens t in the concatenated sequence (xp​,xr​), making it explicitly token-wise. The same applies to $L_{adv}$.
> >
> > This ensures the model preserves the exact token-wise distribution of the reference model on safe text, preventing the "catastrophic forgetting" often seen in unlearning.
> >
> > **Motivation: Why does "indistinguishability" in the representations ensure safety?**
> > If the latent representation of a toxic continuation contains no information distinguishing it from a benign continuation, the subsequent decoding head (which acts on this representation) cannot generate the toxic content. Unlike RMU (which maps toxic representations to random noise), or Circuit Breakers (which orthogonalizes them), REPO maps toxic representations toward their specific semantic antonyms (the benign pairs). This preserves the semantic structure of the model while surgically removing the specific "direction" that encodes toxicity.
> >
> > The key novelty of REPO compared to RMU and CB is leveraging the availability of both positive and negative continuations for each prompt in the detoxification task. The retain loss ensures that positive

---

### Meta-Review · Area_Chair_EMyC · 2026-01-07

**Summary:**

Out of three, two reviewers gave a score of 2 with major fundamental concerns relating the central claim, formulation, novelty, and the experiments. The reviewer who gave a score of 6 also had major concerns (and was very less confident compared to the other two).

**Reviewer Concerns:**

I think this paper requires major revision. Please see my comment above.

**Reviewer Scores:**

I don't think the scores would have changed enough to allow acceptance of the paper.

---

### Decision · Program_Chairs · 2026-01-26

Reject